# Risk Factors for Antimicrobial Resistance in Turkey Farms: A Cross-Sectional Study in Three European Countries

**DOI:** 10.3390/antibiotics10070820

**Published:** 2021-07-06

**Authors:** Mayu Horie, Dongsheng Yang, Philip Joosten, Patrick Munk, Katharina Wadepohl, Claire Chauvin, Gabriel Moyano, Magdalena Skarżyńska, Jeroen Dewulf, Frank M. Aarestrup, Thomas Blaha, Pascal Sanders, Bruno Gonzalez-Zorn, Dariusz Wasyl, Jaap A. Wagenaar, Dick Heederik, Dik Mevius, Heike Schmitt, Lidwien A. M. Smit, Liese Van Gompel

**Affiliations:** 1Institute for Risk Assessment Sciences, Utrecht University, Yalelaan 2, 3584 CM Utrecht, The Netherlands; d.yang@uu.nl (D.Y.); d.heederik@uu.nl (D.H.); h.schmitt@uu.nl (H.S.); l.a.smit@uu.nl (L.A.M.S.); l.vangompel@uu.nl (L.V.G.); 2Veterinary Epidemiology Unit, Department of Obstetrics, Reproduction and Herd Health, Faculty of Veterinary Medicine, Ghent University, Salisburylaan 133, 9820 Merelbeke, Belgium; philip.joosten@ugent.be (P.J.); jeroen.dewulf@ugent.be (J.D.); 3Research Group for Genomic Epidemiology, The National Food Institute, Technical University of Denmark, Kemitorvet, 2800 Kgs. Lyngby, Denmark; pmun@food.dtu.dk (P.M.); fmaa@food.dtu.dk (F.M.A.); 4Field Station for Epidemiology, University of Veterinary Medicine Hannover, Büscheler Straße 9, 49456 Bakum, Germany; katharina.wadepohl@tiho-hannover.de (K.W.); thomas.blaha@tiho-hannover.de (T.B.); 5Epidemiology, Health and Welfare Unit, The French Agency for Food, Environmental and Occupational Health & Safety (ANSES), 22440 Ploufragan, France; claire.chauvin@anses.fr (C.C.); pascal.sanders@anses.fr (P.S.); 6Antimicrobial Resistance Unit (ARU), Animal Health Departement, Faculty of Veterinary Medicine and VISAVET Health Surveillance Centre, Complutense University of Madrid, 28040 Madrid, Spain; gmoyano@ucm.es (G.M.); bgzorn@ucm.es (B.G.-Z.); 7Department of Microbiology, National Veterinary Research Institute (PIWet), Partyzantów Avenue 57, 24-100 Puławy, Poland; magdalena.skarzynska@piwet.pulawy.pl (M.S.); wasyl@piwet.pulawy.pl (D.W.); 8Department of Infectious Diseases and Immunology, Faculty of Veterinary Medicine, Utrecht University, Yalelaan 1, 3584 CL Utrecht, The Netherlands; j.wagenaar@uu.nl (J.A.W.); d.j.mevius@uu.nl (D.M.); 9Department of Bacteriology and Epidemiology, Wageningen Bioveterinary Research, Houtribweg 39, 8221 RA Lelystad, The Netherlands; 10National Institute for Public Health and the Environment, P.O. Box 1, 3720 BA Bilthoven, The Netherlands

**Keywords:** antimicrobial use, antimicrobial resistance, turkeys, poultry, farm, antimicrobial resistance genes, biosecurity, risk factor, metagenomics, qPCR, isolates

## Abstract

Food-producing animals are an important reservoir and potential source of transmission of antimicrobial resistance (AMR) to humans. However, research on AMR in turkey farms is limited. This study aimed to identify risk factors for AMR in turkey farms in three European countries (Germany, France, and Spain). Between 2014 and 2016, faecal samples, antimicrobial usage (AMU), and biosecurity information were collected from 60 farms. The level of AMR in faecal samples was quantified in three ways: By measuring the abundance of AMR genes through (i) shotgun metagenomics sequencing (*n* = 60), (ii) quantitative real-time polymerase chain reaction (qPCR) targeting *ermB*, *tetW*, *sul2*, and *aph3′-III*; (*n* = 304), and (iii) by identifying the phenotypic prevalence of AMR in *Escherichia coli* isolates by minimum inhibitory concentrations (MIC) (*n* = 600). The association between AMU or biosecurity and AMR was explored. Significant positive associations were detected between AMU and both genotypic and phenotypic AMR for specific antimicrobial classes. Beta-lactam and colistin resistance (metagenomics sequencing); ampicillin and ciprofloxacin resistance (MIC) were associated with AMU. However, no robust AMU-AMR association was detected by analyzing qPCR targets. In addition, no evidence was found that lower biosecurity increases AMR abundance. Using multiple complementary AMR detection methods added insights into AMU-AMR associations at turkey farms.

## 1. Introduction

Antimicrobial resistance (AMR) is a global public health concern causing a substantial health and economic burden [1]. The types of antimicrobials used in food-producing animals are often the same or closely related to those used in human medicine [2]. Besides, resistance can spread rapidly and unpredictably through various environments. Therefore, AMR developed in animals can also be transferred to humans. To combat this, AMR is being addressed as part of a One Health approach [3,4].

Turkeys and turkey meat are possible sources for the transmission of AMR [5]. Within the European poultry sector, turkey fattening is the second biggest meat production sector after broiler production, accounting for around 14% of overall poultry meat production [6]. Recently, monitoring data in European countries has shown that a substantial proportion of isolates from turkeys are resistant to several classes of antimicrobials [7].

Farm-level risk factors for AMR in turkeys, such as antimicrobial usage (AMU) and biosecurity measures, have been examined in specific countries [8,9,10,11,12,13]. For example, AMU in the flock and evidence of mice were reported as risk factors for ciprofloxacin resistance in *Escherichia coli* (*E. coli*) in Great Britain [8]. In Germany, the floor design of turkey houses did not affect the development of resistance to enrofloxacin and ampicillin in *E. coli* isolates from turkeys [12,13]. However, it is unclear if these risk factors are country specific or not, because large variation exists between countries and farms in terms of the amount and type of antimicrobials used [14]. Furthermore, farming practices, including biosecurity measures, vary between countries and farms. Therefore, risk factors for AMR at a regional level may not be predictive for other regions or countries.

So far, all studies in turkeys have focused on the prevalence and characteristics of phenotypic resistance. Bacterial species such as *E. coli*, *Salmonella enterica*, and *Campylobactor* spp. were isolated from faeces and minimum inhibitory concentrations (MIC) were determined for fixed panels of antimicrobials [8,9,10,11,12,13,14,15]. There are many mechanisms by which these specific bacteria acquire resistance to antimicrobials. For example, there are multiple gene families encoding extended spectrum beta-lactamases (ESBL) or plasmid-mediated AmpC beta-lactamases. The enterobacteriaceae producing these enzymes are resistant to antibiotics such as penicillins and 3rd and 4th generation cephalosporins. These isolates can then transfer ESBL or AmpC genes to other bacteria in the gut environment or through the food chain. In poultry production pyramids, ESBLs are frequently found [16]. Therefore, culture-dependent methods may underestimate AMR in unculturable gut microbiota. Genotypic methods enable faecal AMR gene detection. When using metagenomics or quantitative real-time polymerase chain reaction (qPCR), the abundance and diversity of AMR genes present in samples can be measured without culturing bacteria. Combining this kind of AMR data with data on AMU and other potential on-farm risk factors, allows for exposure-response relationships to be explored [17,18,19]. Comparing AMR detection methods provides a better understanding of the complex mechanisms behind AMR occurrence in food-producing animals.

As part of the Ecology from Farm to Fork of Microbial Drug Resistance and Transmission (EFFORT) project (http://www.effort-against-amr.eu/, accessed on 28 March 2021), the present study aimed to explore AMR in turkeys from 60 farms in three European countries. The objectives of this paper were to (i) quantify the abundance and diversity of AMR genes in turkey faeces by applying metagenomics and qPCR, and to (ii) determine risk factors for AMR such as AMU as well as other potential farm-level risk factors. In addition, the used AMR quantification methods were compared.

## 2. Results

### 2.1. Overview of the Sampled Farms and Flocks

General characteristics of the sampled farms (*n* = 60) are shown in Table 1. The total number of turkeys per farm varied considerably (median 10,000 turkeys per farm, range: 2950–56,850). We carried out sampling across all seasons: Spring (*n* = 21), summer (*n* = 8), autumn (*n* = 16), and winter (*n* = 15). All farms in country H were sampled in spring and summer. The weight of turkeys at the set up differed substantially between the three countries, and within country B. In country H, all the farms followed an integrated fattening process where the turkeys were introduced to the fattening farms after 28 days of life in breeding, resulting in a small variation in set up weights.

The median age of turkeys at sampling was 115 days. Flocks were separated by sex in country B and H, with the exception of country E where both cocks and hens were usually housed together with a mobile fence. Therefore, some of the hens within those flocks had been removed from the house prior to sampling of the cocks. The overall expected slaughter age was 118 days. For some flocks we could not exactly determine how many days before slaughter sampling was performed, since these included several groups with a different expected slaughter date. Consequently, we calculated the average expected slaughter age per flock.

The biosecurity status at the farm was reduced to two levels. Due to a large number of questions, the questions that were significantly related with AMR in the applied models were shown in Table 1 with the number of farms that answered yes. The proportion of farms that answered yes differed between countries for several biosecurity statuses. For instance, farms where turkeys had outdoor access were only included in country B (70% of the farms in country B).

### 2.2. Antimicrobial Usage

Antimicrobial group treatments applied during the entire rearing period of the sampled flock were quantified using treatment incidence (TI) as a unit of measurement.

There were differences in amounts and types of antimicrobials used between countries (Figure 1). The mean TI per farm was 8.03, 9.95, and 18.4, in country B, E, and H, respectively. Aminoglycosides and spectinomycins, and macrolides and lincomycins were grouped together because they have a common resistance mechanism. The most frequently used antimicrobial groups across all the farms were beta-lactams, polymyxins, and quinolones.

The sum of TI at 60 farms is shown in Appendix A. Across all farms, 7 (11.7%) did not use any antimicrobials (country B:3, E:3, and H:1).

### 2.3. AMR Genes Identified by Metagenomics

#### 2.3.1. The Abundance and Composition of AMR Genes

In total, 573 different AMR genes were identified in samples from 60 turkey farms using ResFinder as a reference database [20]. The abundance of AMR genes were quantified using normalized fragments per kilobase reference per million bacterial fragments (FPKM) values. The FPKM values for the different AMR genes were summed for each antimicrobial class. In general, the composition of AMR genes appeared rather homogenous across farms despite the difference in AMU, and even when comparing farms that did or did not use antimicrobials (Figure 2). The clusters of AMR genes encoding for resistance to tetracyclines, macrolides, and aminoglycosides were most abundant. Moreover, AMR gene clusters encoding for resistance to aminoglycosides, beta-lactams, macrolides, phenicols, sulphonamides, tetracyclines, and trimethoprim classes were detected on all farms. A stacked bar chart showing FPKM values (i.e., not proportional) is shown in Appendix A.

The total abundance of AMR genes, expressed as the summed FPKM values differed between the three countries. The mean total abundance on the farms in country E was significantly lower than that of country H (One-way ANOVA, Tukey HSD, *p* < 0.01) (Appendix A).

#### 2.3.2. Factors Associated with the Abundance of AMR Gene Clusters

Factors associated with the abundance of AMR gene clusters of eight antimicrobial classes were investigated for 57 farms with complete data (country B: *n* = 18, E: *n* = 20, and H: *n* = 19). Using a random-effects meta-analysis by country, Table 2 presents the associations between AMU and the abundance of AMR gene cluster of the corresponding antimicrobial class. Three significant associations between AMU and the corresponding AMR gene cluster were detected: Beta-lactam use (penicillin and aminopenicillins) and beta-lactam resistance, polymyxin use, and colistin resistance, and aminoglycosides or spectinomycin use (binary variable), and aminoglycoside resistance (*p* value < 0.1 adjusted for multiple testing). At farms that reported a higher TI of beta-lactam and polymyxins, a higher faecal abundance of the corresponding AMR gene clusters was observed. Farms with reported aminoglycosides or spectinomycin use had a higher faecal abundance of aminoglycoside resistance genes compared to the farms that did not use these antimicrobial classes. However, only one and five farms reported usage of aminoglycoside and lincomycin-spectinomycin, respectively.

None of the other farm characteristics than AMU were significantly associated with the abundance of AMR gene clusters after Benjamini–Hochberg multiple testing correction (adjusted *p* value ≥ 0.1).

### 2.4. ermB, tetW, sul2, and aph3′-III Identified by qPCR

#### 2.4.1. Abundance of ermB, tetW, sul2, and aph3′-III

In total, 304 samples were analyzed by qPCR. Across all samples, the number of 16S rRNA gene copies varied (log_10_ copies median = 10.8, min = 7.73, and max = 12.8). The number of 16S rRNA copies were used subsequently to calculate relative concentrations of the AMR gene copies. After the qPCR quality check, in order to include samples with a low concentration of *sul2* (11 samples) and *aph3′-III* (20 samples) that were below the limit of detection or limit of quantification, the following values were assigned: *sul2*: 5.10; *aph3′-III*: 3.62. The unit was the number of gene copies (log_10_ copies) before normalization with 16S rRNA. Of those, two *aph3′-III* samples were removed due to a low abundance of 16S rRNA (log_10_ 16S rRNA copies < 8.51). As a result, 283 (93.1%), 287 (94.4%), 262 (86.1%), and 269 (88.5%) samples for *ermB*, *tetW*, *sul2*, and *aph3′-III*, respectively, were available for analysis. The abundance of the four genes relative to bacterial DNA (16S rRNA), stratified per country and gene is shown in Figure 3.

#### 2.4.2. Factors Associated with the Abundance of *ermB*, *tetW*, *sul2*, and *aph3′-III*

In the univariate analysis, total AMU (summed TI of all the antimicrobial classes at farm level) was positively associated with the abundance of *ermB* (Geometric Mean Ratio, GMR = 1.86) and *tetW* (GMR = 1.81). No significant association between AMU and the corresponding resistance gene abundances were detected (Appendix A).

Table 3 presents GMR estimates and 95% confidence intervals of the final multivariable models mutually adjusted for technical farm characteristics and biosecurity. None of the biosecurity variables were associated with the abundance of *sul2*. Linear mixed models with random effect for country were fitted for all the genes, however, there was no variance between countries in the final *sul2* model.

Trimethoprim-sulphonamide treatment in flocks was positively associated with the abundance of *sul2* in turkey faeces, when adjusted for sampling season and the presence of other livestock at the farm (GMR = 7.38). No association was detected between the abundance of *ermB*, *tetW*, and *aph3′-III* and the use of corresponding AMU in multivariable models. Three biosecurity variables remained in the final *ermB* model, and two in the final *tetW* and *aph3′-III* models. The abundance of *ermB* and *tetW* in faeces was significantly lower at farms where visitor access was granted more than once a month, and at farms where turkeys had outdoor access. The concentration of *ermB* in faeces was lower if there were different age categories of turkeys present on the farm. For the abundance of *aph3′-III*, having wild bird- and vermin-proof grids placed on the air inlets was positively associated while having a permanent staff that keeps turkeys or birds at home was negatively associated.

### 2.5. Phenotypic Resistance Identified by Minimum Inhibitory Concentrations

#### 2.5.1. *E. coli* Resistance to Antimicrobials

Ceccarelli et al., previously described the MIC values derived from the turkey faeces collected in this study [21]. *E. coli* was successfully isolated from 596 out of 600 samples, and MIC values were determined by broth microdilution for a fixed panel of 14 antimicrobials for those isolates. Epidemiological cut-off values were used to determine non-wild type susceptible (i.e.microbiological resistant) isolates. However, misinterpretation of sulphamethoxazole MIC-endpoints (overestimation of resistance) for country B led to the exclusion of these data from the analysis.

The proportions of resistant *E. coli* isolates differed between countries and between antimicrobials [21]. The proportion of isolates resistant to ampicillin and tetracycline was higher than 70% in all three countries. The proportion of isolates resistant to ciprofloxacin, nalidixic acid and chloramphenicol was higher than 55% in country H, whereas those in both country B and E were less than 35%. Less than 10% of all the isolates were resistant to cefotaxime, ceftazidime, meropenem, azithromycin, gentamicin, and tigecycline. All meropenem-resistant isolates were confirmed to be negative for known carbapenemases by PCR.

#### 2.5.2. Factors Associated with *E. coli* Resistance

The univariate association between potential risk factors and the occurrence of *E. coli* resistant to ampicillin, tetracycline, and ciprofloxacin from the mixed effect logistic models are presented in Appendix A. These three antimicrobials were selected for this analysis because both (i) the number of farms on which corresponding antimicrobial classes were used and (ii) the prevalence of isolates resistant to the antimicrobials were higher than 10%. Significant positive associations were detected between AMU and the occurrence of *E. coli* resistant to ampicillin, tetracycline, and ciprofloxacin. The total amount of AMU was also positively related to resistance to all three antimicrobials. In addition to these three antimicrobials, a univariate association between polymyxin use and resistance to colistin was detected (*p* = 0.001). However, because of model convergence failure, the multivariable model for colistin resistance could not be investigated. A random intercept for farms was included in all the models and country intercept was also added to the ciprofloxacin model because it significantly improved the model fit.

Table 4 shows that there was a significant positive association between AMU at the farm and resistance of *E. coli* isolates for ampicillin and ciprofloxacin when mutually adjusted for other farm characteristics. The presence of a turkey farm within 500 m was negatively associated with ciprofloxacin resistance of *E. coli* isolates. Other associations between biosecurity and resistance of *E. coli* isolates were not statistically significant after mutual adjustment for potential other determinants identified in the univariate analysis.

### 2.6. Correlations between AMR Genes Abundances Detected by Metagenomics and qPCR

The correlation between an abundance of *ermB*, *tetW*, *sul2*, and *aph3′-III* detected by metagenomics and qPCR is shown in Appendix A. Metagenomics samples were pooled at a farm level and the median of the qPCR samples per farm were used. A significant but modest correlation was observed for all four genes (*p* < 0.001, Spearman rho = 0.47–0.74). The highest correlation was observed for *ermB* (rho = 0.74).

The abundance of metagenomically-derived AMR genes clustered at the 90% identity level and present within the macrolide, tetracycline, sulphonamide, and aminoglycoside class clusters were shown in Appendix A. The abundance of *ermB*, *tetW*, *sul2*, and *aph3′-III* accounted for 69.0%, 42.3%, 42.6%, and 25.3% of the macrolide, tetracycline, sulphonamide, and aminoglycoside resistance class clusters, respectively.

## 3. Discussion

In this multi-country risk factor study on 60 turkey farms, we investigated risk factors for the faecal abundance of AMR genes in turkeys detected by both metagenomics and qPCR, as well as the prevalence of resistance in *E. coli* isolates in turkey faeces collected in Germany, France, and Spain. We detected positive associations between AMU and both genotypic and phenotypic AMR, specifically for beta-lactam and colistin resistance (metagenomics) as well as ampicillin and ciprofloxacin resistance (MIC).

Substantial differences in AMU were observed between farms and countries. The most frequently used antimicrobial groups were beta-lactams (aminopenicillins and penicillins), followed by polymyxins, and quinolones (fluoroquinolones and other quinolones). A previous study on Italian turkey farms reported that polymyxins, penicillins (including aminopenicillins), and sulphonamides were widely used [22]. A substantial variation in the use of antimicrobial classes within and between countries is expected since there are many possible explanations such as differences in antimicrobial stewardship of veterinarians, differences in availability of pharmaceutical products, and national legislations [23]. A similar high variation in AMU was observed on broiler farms from nine European countries [24].

The relative AMR gene composition detected by metagenomics was similar across the 60 included farms, including flocks that did not receive any antimicrobial treatment. This was in accordance with European broiler studies, where the faecal AMR genes composition appeared to be roughly similar between farms, despite the absence of AMU in many flocks [18,25]. Genes encoding for resistance to tetracyclines were the most dominant cluster, followed by macrolides and aminoglycosides, when clustered at the antimicrobial class level. This is consistent with previously published gut microbiome data in Polish turkeys [26]. These classes, however, did not correspond with the most frequently used antimicrobials in our study. The presence of these AMR gene classes in the faeces of other animal species is reported in multiple countries, regardless of AMU [25,26,27]. These AMR genes may be present in various bacterial species in the gut of turkeys. It suggests that there are other factors that affect the composition of AMR genes in the gut environment, in addition to direct AMU. This could include the co-selection of resistance by AMU in the production round or in previous rounds at the farm, through which antimicrobial residues and resistant bacteria remained in the environment. The physical transfer of bacteria via the movement of animals may have contributed as well [28].

Significant positive associations were detected between AMU and the abundance of corresponding AMR genes for some antimicrobial classes. The result of the random effects meta-analyses using metagenomics data showed that flocks that received more beta-lactam and colistin antimicrobials had a higher abundance of the corresponding AMR genes. Horizontal gene transfer plays a role in the acquisition of beta-lactam and colistin resistance in addition to chromosomal mutations [29,30]. Therefore, AMU may select for and thus accelerate such transmission.

Fluoroquinolone use has previously been identified as a risk factor for increased fluoroquinolone resistance in *E. coli* [9,12,13]. These studies also reported an increased prevalence of ampicillin resistant isolates in trials in the absence of ampicillin use [12,13]. In line with these studies, we also observed that increased fluoroquinolone use was related to higher proportions of *E. coli* isolates being resistant to ciprofloxacin. In addition, we observed an AMU-AMR association for ampicillin in *E. coli* isolates. Boulianne et al. reported associations between tetracycline use and the occurrence of tetracycline resistance in *E. coli* isolates in Canadian turkey flocks [11]. We also observed positive phenotypic AMU-AMR associations on tetracycline in our study, which were statistically significant in the univariate analysis, but with a wide confidence interval. To study more phenotypic AMU-AMR associations, susceptibility testing in gram positive bacteria such as *Enterococcus* spp. could be considered [11].

We found no evidence that good biosecurity measures were related to lower faecal AMR abundance in turkeys. Our results differ from earlier findings on the association between biosecurity measures and fluoroquinolone resistance in *E. coli* in turkey faeces in Great Britain [8]. They reported that the on-farm presence of mice was a risk factor, while disinfection of floors and walls at depopulation appeared protective. However, information on the quantity of AMU in the sampled flock was not included in their study, so it may be possible that AMU was correlated with the biosecurity factors. In our study, we could not verify if the presence of mice increases the risk, but we observed that bird- and vermin-proof grids placed on the air inlets were associated with a higher risk for *aph3′-III* detected by qPCR. Additionally, the fact that all the farms provided the same answer for “there is a preventive vermin control program” and “stables are disinfected after every round” in our study may suggest that these measures are not associated with variations of AMR on turkey farms. Chuppava et al. reported that the floor design of the turkey house did not correlate with the development of ampicillin- and enrofloxacin-resistant *E. coli* isolates [12,13]. Furthermore, there was little evidence for associations between farm biosecurity and the abundance of AMR genes in European broilers [18]. Interestingly, poor biosecurity such as staff having contact with other birds among others, were in fact related to a lower faecal abundance of *aph3′-III* detected by qPCR. In addition, the presence of a turkey farm within 500 m was negatively associated with *E. coli* resistance to ciprofloxacin. However, we cannot explain this phenomenon biologically. Therefore, the relationship between biosecurity and AMR on turkey farms remains uncertain.

Three different AMR detection methods were used in this study. We observed modest correlations between the abundance of AMR genes quantified by metagenomics and qPCR. A possible reason may be the difference in sample selection. For metagenomic sequencing, the samples were pooled per farm before DNA extraction to represent the farm, whereas DNA was extracted from five to six samples individually for qPCR analysis to detect variations within farms. Pooled samples provide a composition representative of the common AMR genes at the farm [31], whereas the abundance of particular genes may vary between individual samples. Additionally, a low correlation could be due to the low concentration of the target genes or inhibition of gene expression [32]. We chose multiple genes in metagenomic sequencing based on 90% identity level and summed to compare with qPCR, but we can also speculate that there might have been more genes that qPCR detects. On the other hand, the agreement between the abundance of genotypic resistance and phenotypic resistance was not tested. This is because genotypic resistance in this study represents the abundance in the total faecal bacterial community whereas phenotypic resistance is specific to *E. coli*. To compare and predict phenotypic resistance in specific isolates, whole genome sequencing studies could be performed [33].

Detecting total genotypic resistance in samples, rather than isolating specific bacteria, is a good choice to find risk factors for AMR genes associated with horizontal gene transfer. Genotypic detection methods in our study enabled to confirm that AMR genes were widely present in turkey faeces for some antimicrobial classes such as macrolides and aminoglycosides, despite low phenotypic resistance to specific antimicrobials expressed in *E. coli*. The strength of metagenomic sequencing was that it showed the composition of AMR genes in the resistome (the collection of all resistance genes in a sample). Moreover, AMR genes could be analyzed at several grouping levels, such as at a gene and antimicrobial class level. On the other hand, qPCR may be a better choice for detecting specific genes of interest because of lower costs and simple procedures over metagenomic sequencing. However, the selection of the most appropriate target gene may be difficult. Our qPCR target genes were the most abundant gene clusters within the respective antimicrobial classes. However, such information may not always be available beforehand. Limitations for both metagenomic sequencing and qPCR lie in the difficulty to compare the results with other studies since genotypic AMR data in turkeys is still scarce and methods can vary between studies. In contrast, phenotypic AMR in specific bacteria has been studied in a standardized manner for monitoring purposes, making it easier to compare results between studies or to monitor trends over time. However, using dichotomized outcomes by epidemiological cut-off in our study hampered data analysis for antimicrobial classes in which the resistant proportion of isolates were low. In summary, we showed that these methods are complementary and the choice depends on the research question.

Our study is unique considering that farms were included from three European countries using standardized sampling, which enabled the identification of risk factors that are not country-specific. We also related AMU and multiple farm-level factors to both genotypic and phenotypic AMR. However, information on purchased AMU at a farm level was not available in all countries and could therefore not be studied as an alternative to group treatments. This could explain the on-farm background levels of AMR in the absence of reported usage. Moreover, although we included group treatments data at breeding farms, farm characteristics of those farms were not collected. Both AMU and biosecurity information of the sampled farms were from farmers’ reports rather than registered data. Therefore, underreporting of AMU and misclassification or missing biosecurity answers could have led to social desirability bias. We quantified the 16S rRNA gene to normalize AMR gene results detected by qPCR, but many bacterial species have more than one copy of the 16S rRNA gene. There is no suitable approach to correct for copy numbers in microbiome data [34,35]. Although gut bacterial composition between turkeys may differ, we expect that this taxonomic difference will not have a large effect on the between group comparisons. Error in quantification of the 16S rRNA gene that we used to normalize the AMR genes would lead to a less precise estimate of AMR, resulting in the attenuation of risk estimates (e.g., AMU-AMR associations). Despite these limitations, our study shows an association between AMU and AMR on turkey farms, which is a potential exposure route to humans.

## 4. Materials and Methods

### 4.1. Selection of Farms

Between October 2014 and October 2016, 60 conventional fattening turkey farms were visited in 3 countries (Germany, France, and Spain, 20 farms per country). German farms were geographically spread over the country, while all French and Spanish farms were concentrated in Brittany and Andalusia, respectively, both being the major turkey production sites of these countries (Appendix A). The preferable selection of farms was based on the following criteria: Conventional farms with an all-in-all-out system and containing 3000–15,000 birds per farm. However, the size criteria were not always met. Farms included in the study were unrelated. Both farms and countries were anonymized (country B, E, and H) to ensure that the results cannot be traced back, consistent with previous EFFORT publications in which data from 9 countries (A to I) was analyzed. The selected farms cannot be considered representative for the respective countries.

Each farm was visited once to collect faecal samples. On each farm, the unit for sampling was a turkey house with a flock that had not been moved or mixed with other flocks except the removal of individual birds before the sampling time. In the flocks, all animals had received the same group treatments by water, medicated feed, or injection during their lifetime. The sampling was intended at maximally one week before the final slaughter date of the hens, but samples were collected randomly regardless of sex. Farms were visited across all seasons.

### 4.2. Questionnaire Data: Antimicrobial Usage, Farm Characteristics, and Biosecurity

Information on all antimicrobials administered as group treatments to the sampled flock during their whole lifetime were documented by the farmers with the supervision of the researchers or veterinarians. Before introducing the sampled flock, researchers informed the farmers on how to document the antimicrobial treatments. Group treatment data included not only those administered in the sampled farms but also in previous breeding farms if applicable. Technical farm characteristics and biosecurity status were obtained by a questionnaire filled out by the participating farmers. Answers in the questionnaire were entered into EpiData version 3.1 Software (EpiData Association, Odense, Denmark).

### 4.3. Quantification of AMU

To quantify AMU, TI was calculated based on antimicrobials administered to the sampled flock, as previously described [21,24]. Defined Daily Dose for turkeys (DDD_turkey_) were assigned for all the antimicrobials used on the included farms. Therefore, TI is expressed as the number of DDD_turkey_ administered per 100 turkey days at risk or the number of days per 100 turkey days that the flock received a standardized dose of antimicrobials (1). The latter can also be interpreted as the percentage of time that a turkey is treated with antimicrobials in its life:(1)$$\mathrm{TI}=\frac{Total\;amount\;of\;active\;substance\;administered\;\left(mg\right)}{DDD\;turkey\;\left(mg/kg/day\right)\times number\;of\;days\;at\;risk\times kg\;turkey\;at\;risk}\times 100\;turkeys\;at\;risk$$

For determining “kg turkey at risk”, a standard weight of 6 kg was used according to the European Surveillance of Veterinary Antimicrobial Consumption (ESVAC) guidelines [36]. Then, the standard weight was multiplied by the number of turkeys at setup. “Number of days at risk” was equal to the expected age of slaughter at each farm. When there were a few different age groups of slaughter batches within the sampled flock, the average age within the sampled flock was used. From this formula, TI was calculated for each antimicrobial class per farm. Total TI per farm was also calculated.

For the risk factor analyses, the sum of TI at farm level for each antimicrobial class was used. Furthermore, we grouped antimicrobials (TIs) that possessed similar mechanisms of resistance, i.e., macrolides and lincomycin, aminoglycosides and spectinomycin. Since lincomycin and spectinomycin were administered as combination products with a fixed ratio (lincomycin:spectinomycin, 1:2) [37], TI was first calculated using DDD _turkey_ for lincomycin-spectinomycin and subsequently divided for each active substance. Aminopenicillin and penicillin were grouped as beta-lactam, fluoroquinolones, and other quinolones (flumequine) were grouped together as quinolones.

### 4.4. Sampling and Processing of Faecal Samples

Per farm, 25 fresh faecal droppings were collected from the floor of one turkey house. After collection, each sample was refrigerated at 4 °C and transported to the laboratories within 24 h.

On arrival at the labolatory of each sampling country, samples for *E. coli* isolation were processed. Simultaneously, samples for metagenomics and qPCR were prepared and stored at −80 °C until shipment. Frozen samples were shipped on dry ice to the Institute for Risk Assessment Sciences (IRAS, Utrecht, the Netherlands).

### 4.5. Metagenomic Sequencing and Processing Data

Metagenomic sequencing and processing was performed as described previously, with modifications [25,38]. The reads are available in the European Nucleotide Archive, under project accession number PRJEB39685.

At the laboratory, the individual faecal samples were homogenized by stirring thoroughly with a tongue depressor or a spoon for a few minutes. Twenty-five individual samples from the same farm were pooled with 0.5 g of faeces from each sample and stirred for a few minutes. DNA extraction was centrally performed at the Technical University of Denmark (The National Food institute, Kgs. Lyngby, Denmark). From a 0.2-g sample, DNA was extracted using a modified QIAmp Fast DNA Stool Mini Kit (QIAGEN, Hilden, Germany) [39]. The samples were sequenced on the NovaSeq 6000 platform (Illumina Inc, CA, USA) by Admera Health (South Plainfield, NJ, USA), using a 2 × 150-bp paired-end (PE) read approach, aiming for 35 M PE reads per sample.

After removing low-quality nucleotides as well as adaptor sequences, trimmed read pairs corresponding to each farm-level sample were aligned to the ResFinder database and, separately, to a merged database of genomic sequences using the k-mer alignment software KMA (v1.2.8). The ResFinder database repository was accessed on 13 February 2019, and contained 3081 AMR genes. Read was aligned to the ResFinder database using the KMA parameters ‘-mem_mode -ef -1t1 -cge -nf -nc’. In order to filter out low-coverage alignments, alignments that were lower than a 20% consensus of the corresponding reference were removed. The genomic sequence database was described previously [40]. Reads were assigned to the the genomic database using KMA parameters ‘-mem_mode -ef -1t1 -apm f -nf -nc’. The sum of sequencing fragments mapped to the bacteria, archaea, plasmid, bacteria_draft, HumanMicrobiome, and MetaHitAssembly sub-databases was used as the sample size factor for the FPKM calculation.

As the unit of outcome, FPKM values were computed as previously described [25]. The values were aggregated at the antimicrobial class cluster level for risk factor analysis. Distribution was checked and a pseudocount of one and log_10_ transformation was applied to FPKM values. Furthermore, the values were aggregated at the 90% identity clustering [41], to analyze the abundance of the specific AMR genes.

### 4.6. qPCR Analysis

For qPCR analysis, 5 to 6 samples per farm were randomly selected, resulting in 304 samples. Five samples per farm were incldued to depict between-animal variation which is assumed to be small within one turkey house. From each sample, 0.5 g of faeces were transferred to a 2-mL cryotube. From a 0.2-g sample, DNA was extracted using a modified QIAmp Fast DNA Stool Mini Kit (QIAGEN, Hilden, Germany) [39]. For all the samples, DNA extraction was performed centrally at IRAS, in the Netherlands.

Four AMR genes, *ermB*, *tetW*, *sul2*, and *aph3′-III,* were selected as qPCR targets. These genes encode resistance against macrolides, tetracyclines, sulphonamides, and aminoglycosides, respectively. These antibiotic classes of public health relevance were chosen based on their abundance in metagenomic data of pooled pig and broiler faeces samples collected within the EFFORT project [25]. In addition, 16S rRNA was targeted for normalization of the AMR genes to bacterial DNA in each sample. Three gene targets of qPCR assay (16S rRNA, *sul2*, and *aph3′-III*) were performed at the National Veterinary Institute (PIWet, Puławy, Poland), while the other two (*ermB* and *tetW*) were at IRAS. Overall results were centrally analyzed at IRAS.

A qPCR assay was performed as previously described [*ermB*, *tetW*, 16S rRNA [42]; *sul2* and *aph3′-III* [19]]. Briefly, all samples were run in two technical PCR duplicates with a non-competitive internal amplification control (IAC) to control quality. From raw amplification data, Ct values were derived by the R project package “chipPCR” [43]. For each gene, the number of copies derived from the Ct values were normalized to bacterial load (log_10_ (copies of AMR gene/copies 16S rRNA)).

Among the samples passing the qPCR quality criteria (IAC and replicate consistency), those without a quantifiable 16S rRNA concentration were excluded from further analysis (14 samples). Additionally, *sul2* (11 samples) and *aph3′-III* (20 samples) were below the limit of detection or limit of quantification. Those samples were assigned a value (in log_10_ copies) corresponding to the 1st percentile of the distribution when considering all values of all samples together per gene (*sul2*: 5.10; *aph3′-III*: 3.62). Of those, the samples with a low abundance of 16S rRNA (lower than the 1st percentile of the copy unit of all 16S rRNA concentrations) were excluded from data analyses because these present very high normalized values.

### 4.7. E. coli Isolation and MIC Determination

Isolation of *E. coli* and MIC determination was performed as previously described [21]. The individual samples were suspended in buffered peptone water 1/10 (*w*/*v*) with 20% glycerol in a 2-mL cryotube and thoroughly mixed. Ten samples from each farm were selected (no. 2, 4, 6, 8, 10, 12, 14, 16, 18, and 20), resulting in 600 samples for *E. coli* isolation. Briefly, all samples were inoculated on MacConkey agar and after incubating 24 h overnight, suspected colonies were isolated and confirmed as *E. coli*. Isolated samples were stored individually in buffered peptone water with 20% glycerol at −80 °C. Next, MIC values by broth microdilution were determined for a fixed panel of antimicrobials by commercially-available microtitre plates (Sensititre, EUVSEC, Themo Fisher Scientific UK Ltd., Loughborough, UK). The European Committee on Antimicrobial Susceptibility Testing (EUCAST) epidemiological cut-off values were used to differentiate between wild type and non-wild-type susceptibility.

### 4.8. Variable Selection and Statistical Analysis

First, to examine the association between AMR and farm level factors, univariate models with AMR, and the corresponding AMU were applied, as well as with other farm-level variables selected from the questionnaires. Next, according to the association observed in univariate models, multivariable models were built.

All statistical analyses were performed in R version 3.6.1 (https://www.R-project.org, accessed on 28 March 2021).

#### 4.8.1. Explanatory Variables

The distributions of continuous variables (i.e., AMU, “total number of turkeys at the farm”, “age of turkeys at sampling”) were explored and log_10_ transformed in case of skewness. Age of turkeys was standardized by subtracting the mean and dividing it by the standard deviation to avoid modeling errors due to scale differences between variables. As only a limited number of farms (<10) used trimethoprim-sulphonamide, aminoglycosides, or spectinomycin, we dichotomized these variables. From the questionnaires, the most important farm characteristics variables were selected based on expert knowledge and prior studies [8,17,19,44,45].

In the case of a high correlation between technical farm characteristics and biosecurity variables (Spearman ρ > 0.7), technical farm characteristics variables were selected. Variables without contrast and those with missing values were excluded. One missing value of age of turkeys in country B was replaced with the median age of the sampled birds in country B (134 days). All categorical variables were reduced to two levels to avoid convergence errors in modeling.

Four technical farm characteristics variables, namely, “total number of turkeys at the farm”, “age of turkeys at sampling”, “other livestock is present at the farm”, and “season of the sampling”, as well as 19 biosecurity variables fulfilling the above criteria were considered in the following models (Appendix A).

#### 4.8.2. Factors Associated with AMR Gene Clusters Identified by Metagenomics Sequencing

Three samples from farms for which the metagenomic data could not be matched with the questionnaire data were excluded in the risk factor analysis, resulting in 57 farms to be analyzed (country B: *n* = 18, E: *n* = 20, and H: *n* = 19). The abundance of AMR genes clustered at the antimicrobial class level were used as the outcome variable. Eight clusters with the reported corresponding AMU were chosen for the models. Random effects meta-analyses by country were performed as previously described [17,18]. First, linear regressions were calculated per country, after which the overall associations were calculated using a random effect for country to take the between country variance into account. To prevent certain countries from largely influencing the estimates, the outcome variable were standardized (mean 0, SD 1) by country. R package Metafor was used [46].

Briefly, univariate associations between AMR gene clusters and corresponding AMU, technical farm characteristics, and biosecurity variables were examined. Additionally, the association between the summed FPKM of all the clusters (total FPKM) and total AMU at the farm was analyzed. *p*-values were adjusted for multiple testing by the Benjamini–Hochberg procedure with the false discovery rate set to 10% [47].

#### 4.8.3. Factors Associated with *ermB*, *tetW*, *sul2*, and *aph3′-III* Identified by qPCR

The abundances of the four genes were averaged at the farm level using the median value of the five to six samples within each farm to remove correlation in the farms (i.e., 60 samples in total), instead of adding a random effect for the farms. Random effect for both the farm and country resulted in convergence errors when modeling. Linear mixed models with random intercept for each country were applied for both univariate and multivariable analyses.

First, univariate models were built for each gene to look for factors significantly influencing the AMR gene concentrations. Subsequently, we applied the step function of the R lmerTest package, which performs a backward elimination of non-significant effects in multivariable models [48]. We applied this to the fixed effects while keeping the random effect for country. The variables included in the full models were: (i) The corresponding AMU variable, (ii) the variables significantly related with AMR in the univariate analysis (Satterthwaite’s degrees of freedom method, *p* value < 0.05), and (iii) four technical farm characteristics variables because these may be related with AMU and biosecurity variables. Fixed effect variables were eliminated backward from the full models according to the *p* value (alpha = 0.05), while keeping the corresponding AMU variable. To make the model coefficients more interpretable, all estimates and their 95% CIs were expressed as GMR values by exponentiating with base 10 coefficients (Table 3, Appendix A).

#### 4.8.4. Factors Associated with *E. coli* Resistance

The occurrence of *E. coli* isolates resistant to ampicillin, tetracycline, and ciprofloxacin were used as the outcome variables. These three antimicrobials were selected because there were more than six farms (i.e., 10% of all the farms) with the reported corresponding AMU and there were more than 60 resistant isolates (i.e., 10% of all the isolates). Nalidixic acid was not selected but ciprofloxacin was selected for quinolone resistance. This is because when using the epidemiological cut-off to define non-wild type susceptible isolates, nalidixic acid and the fluoroquinolone ciprofloxacin show the same results in proportions of non-wild type strains. Corresponding AMU variables were aminopenicillin, tetracycline, and quinolone use (fluoroquinolone and other quinolones). Penicillins were not included since *E. coli* is intrinsically resistant to penicillin. At first, it was intended to investigate the association between polymyxin use and colistin-resistant *E. coli*, but many models failed to converge in univariate analysis, which made it impossible to further investigate risk factors. Mixed effects logistic models with random intercept for farm were applied. A country random intercept was added when it improved the fit in null models.

Following univariate analysis, the variables significantly related in univariate analysis (*p* value < 0.05) were added in the multivariable models. All ORs and their 95% CIs are shown in the results (Table 4, Appendix A).

#### 4.8.5. Comparisons between Metagenomics and qPCR

First, two genotypic resistance methods, namely metagenomics and qPCR samples were compared. Metagenomics samples were pooled at the farm level while for qPCR samples, the median value of the five to six samples within each farm were used. Associations between the abundance of *ermB*, *tetW*, *sul2*, and *aph3′-III* clusters as identified by metagenomics and the abundance of these genes by qPCR were examined by calculating the Spearman correlation coefficient (Appendix A). In addition, total abundance (i.e., summed FPKM of all the farms) per gene level cluster was calculated and the proportion of the respective gene within the according macrolide, tetracycline, sulphonamide, and aminoglycoside class level cluster was calculated (Appendix A).

## 5. Conclusions

We investigated risk factors for AMR in European turkey farms using three different AMR detection methods. Positive AMU-AMR associations were detected for both genotypic and phenotypic AMR: Beta-lactam and colistin (metagenomic sequencing) and aminopenicillin and fluoroquinolone (MIC). No robust AMU-AMR association was detected by analyzing qPCR targets. No evidence was found that lower biosecurity increases AMR abundance. We showed AMR genes encoding for some antimicrobial classes were abundant in faeces despite the low prevalence of phenotypic resistance in *E. coli* isolates. Since different AMR detection methods provide information on different aspects of AMR, the choice depends on the availability of resources and research questions. We have shown that using multiple complementary AMR detection methods adds insights into AMU-AMR associations in turkey farms.

## Figures and Tables

**Figure 1 antibiotics-10-00820-f001:**
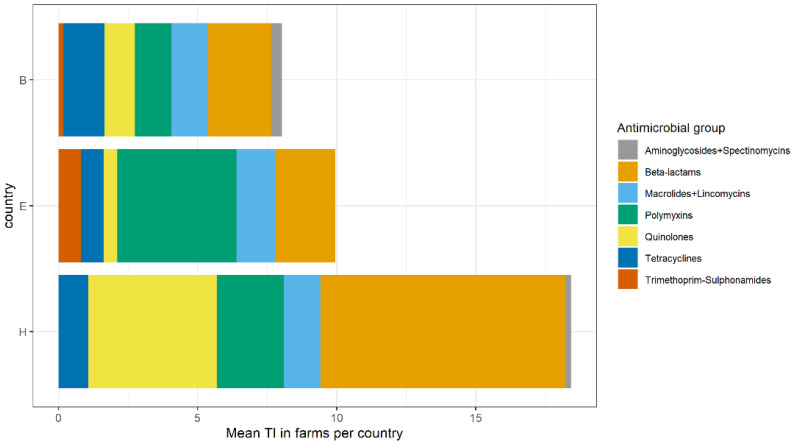
Average antimicrobial usage on farm level in 60 turkey farms in three countries. Mean treatment incidence (TI) shows the average number of treatment days per 100 days. Antimicrobials were grouped after TI was calculated for lincomycin-spectinomycin combination product and subsequently divided and added to macrolides and aminoglycosides, respectively. Beta-lactams included aminopenicillins and penicillins. Quinolones included fluoroquinolones and other quinolones (flumequine). Countries were anonymized as B, E, and H.

**Figure 2 antibiotics-10-00820-f002:**
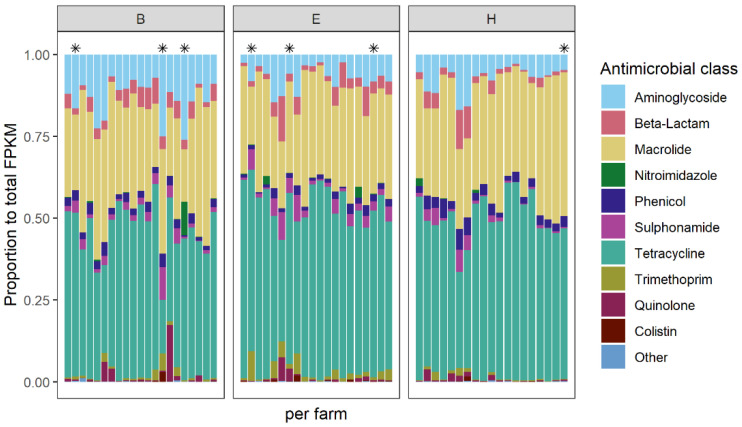
Relative abundance of antimicrobial (AMR) genes expressed as a proportion of total fragments per kilobase reference per million bacterial fragments (FPKM). Columns represent 60 samples from 60 farms from three countries (B: *n* = 21, E: *n* = 20, and H: *n* = 19). One additional farm was visited in country B due to incomplete questionnaire data in one of the farms, resulting in 21 samples in total. One sample in country H was removed due to errors during processing. The AMR genes were aggregated to antimicrobial classes. Seven farms where no antimicrobial use was reported in the sampled flock are indicated with an asterisk above the columns.

**Figure 3 antibiotics-10-00820-f003:**
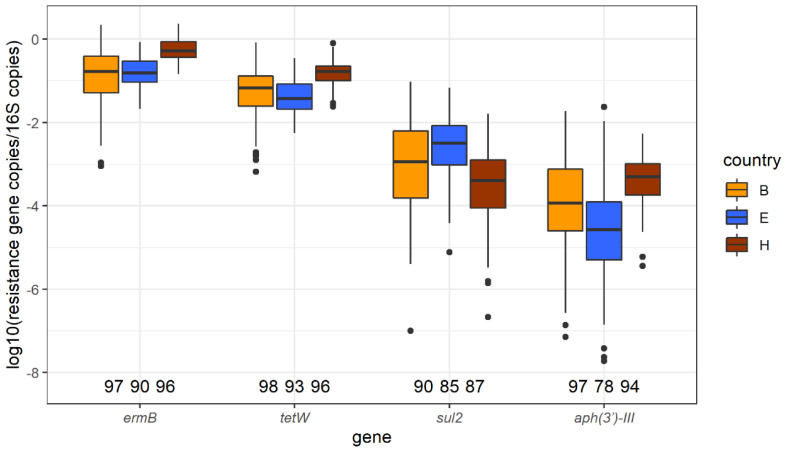
Relative abundance of *ermB*, *tetW*, *sul2*, *and aph3′-III* in turkey faeces sampled in three countries, detected by qPCR. Resistance gene log_10_ copies were normalized using 16S rRNA abundances. The numbers displayed above the horizontal axis are the number of the samples eligible for analysis.

**Table 1 antibiotics-10-00820-t001:** Characteristics of the sampled turkey farms and flocks by country and overall countries.

Characteristics	Country			Overall
	B	E	H	
**Farm Information**				
Included farms, *n*	20	20	20	60
No. of turkeys present on the farm, median (Min-Max)	12,683 (5000–46,500)	7275 (2950–38,000)	12,609 (4404–56,850)	10,055 (2950–56,850)
Farms where other livestock is present, *n* (%)	4 (20)	11 (55)	4 (20)	19 (32)
No. of people working at the farm, median (Min-Max)	2 (1–28)	1.5 (1–3)	1 (1–4)	1.5 (1–28)
Farms sampled in spring and summer, *n* (%)	4 (20)	5 (25)	20 (100)	29 (48)
**Flock Information**				
No. of turkeys at sampling, median (Min-Max) ^a^	4213 (2050–11,660)	4140 (450–9155)	6422 (302–21,356)	4710 (450–21,356)
No. of turkeys at set-up in the current round in the sampled house, median (Min-Max) ^b^	5040 (2997–13,000)	9180 (4240–22,000)	7020 (3000–21,794)	7850 (2997–22,000)
Weight of turkeys at set-up, kg, median (Min-Max) ^c^	1.5 (0.1–6.4)	0.1 (0.1–0.5)	1.1 (0.9–1.3)	1.1 (0.1–6.4)
Age of turkeys at sampling, days, median (Min-Max) ^b^	134 (96–147)	116 (74–140)	101 (86–118)	115 (74–147)
Average expected age at delivery to slaughter, days, median (Min-Max) ^b^	146 (106–154)	109 (79–138)	117 (95–127)	118 (79–154)
**Biosecurity at the Farm**				
Visitor access more than once a month (family members, technicians, etc), *n* (%)	8 (40)	20 (100)	16 (80)	44 (73)
Outdoor access possible for turkeys, *n* (%)	14 (70)	0 (0)	0 (0)	14 (23)
Different age categories of turkeys present, *n* (%)	10 (50)	5 (25)	0 (0)	15 (25)
Bird- and vermin-proof grids placed on the air inlets, *n* (%)	20 (100)	15 (75)	18 (90)	53 (88)
Staff keeps turkeys or birds at home, *n* (%)	2 (10)	7 (35)	1 (5)	10 (17)
Disinfecting footbaths present on the farm, *n* (%)	14 (70)	10 (50)	10 (50)	34 (57)
The nearest turkey farm within 500 m, *n* (%)	4 (20)	5 (25)	4 (20)	13 (22)
Other livestock farm present within 500 m, *n* (%)	12 (60)	18 (90)	7 (35)	37 (62)
Wild birds can enter the stables, *n* (%)	1 (5)	6 (30)	8 (40)	15 (25)

Missing observations were excluded to calculate the average. ^a,b,c^ The number of farms with missing observations: ^a^ 2, ^b^ 1, ^c^ 10. Biosecurity status displayed in the table are those significantly associated with the AMR in the applied models.

**Table 2 antibiotics-10-00820-t002:** Associations between antimicrobial usage (AMU) and relative abundance of the corresponding antimicrobial resistance (AMR) genes detected by metagenomics, obtained from a random-effects meta-analysis by country.

AMU	AMR Gene Cluster ^a^	Estimate	Adjusted *p* Value ^b^	95% CI	Country and Number of Farms with Reported AMU
Log_10_ TI beta-lactam	Beta-lactam	**1.06**	**0.033**	**[0.29–1.84]**	B-15, E-14, H-18
Log_10_ TI polymixin	Colistin	**0.99**	**0.033**	**[0.29–1.69]**	B-4, E-11, H-5
Aminoglycosides or spectinomycin used (ref:no)	Aminoglycoside	**0.92**	**0.097**	**[0.08–1.76]**	B-3, H-3
Trimethoprim-sulphonamides used (ref:no)	Trimethoprim	0.78	0.221	[−0.15–1.71]	B-2, E-3
Trimethoprim-sulphonamides used (ref:no)	Sulphonamide	0.68	0.282	[−0.26–1.61]	B-2, E-3
Log_10_ TI quinolone	Quinolone	0.69	0.338	[−0.43–1.81]	B-5, E-4, H-12
Log_10_ TI tetracyclines	Tetracycline	0.09	0.948	[−0.82–1.00]	B-6, E-6, H-9
Log_10_ TI macrolides + lincomycin	Macrolide	−0.17	0.948	[−1.35–1.01]	B-6, E-12, H-7
Log_10_ TI total AMU	Total FPKM	−0.02	0.948	[−0.62–0.58]	B-15, E-17, H-18

Associations in bold have an adjusted *p* value < 0.1. In the models, 57 farms with complete data were included (country B: *n* = 18, E: *n* = 20, and H: *n* = 19). AMU = Antimicrobial usage; AMR = Antimicrobial resistance; 95% CI = 95% Confidence interval; TI = Treatment incidence. ^a^: Relative abundance of AMR genes were clustered per antimicrobial class and calculated as a sum of fragments per kilobase reference per million bacterial fragments. ^b^: *p* values were adjusted with Benjamini–Hochberg correction with a false discovery rate set to 10%.

**Table 3 antibiotics-10-00820-t003:** Multivariable model associations between antimicrobial usage (AMU), characteristics, biosecurity measures of the turkey farms, and the median relative faecal abundance of *ermB*, *tetW*, *sul2*, and *aph3′-III* per farm.

Model Variables	*ermB*		*tetW*		*sul2*		*aph3′-III*	
	GMR	[95% CI]	GMR	[95% CI]	GMR	[95% CI]	GMR	[95% CI]
**AMU**								
Log_10_ TI macrolides + lincomycin	1.57	[0.77, 3.23]						
Log_10_ TI tetracyclines			1.54	[0.80, 2.97]				
Trimethoprim-sulphonamides used (ref:no)					**7.38**	**[1.61, 33.8]**		
Aminoglycosides or spectinomycin used (ref:no)							1.47	[0.42, 5.14]
**Technical farm characteristics**								
Age of turkeys at sampling (standardized)	**0.73**	**[0.54, 0.98]**						
Other livestock present (ref:no)					**2.89**	**[1.17, 7.14]**	**0.38**	**[0.15, 0.95]**
Sampling season (ref: autumn, winter)					**0.21**	**[0.09, 0.48]**		
**Biosecurity**								
Visitor access more than once a month (ref:no)	**0.41**	**[0.22, 0.75]**	**0.36**	**[0.21, 0.60]**				
Outdoor access possible for turkeys (ref:no)	**0.35**	**[0.17, 0.75]**	**0.37**	**[0.19, 0.74]**				
Different age categories of turkeys present (ref:no)	**0.45**	**[0.25, 0.83]**						
Bird- and vermin-proof grids placed on the air inlets (ref:no)							**6.32**	**[1.76, 22.73]**
Staff keeps turkeys or birds at home (ref:no)							**0.27**	**[0.09, 0.83]**

Associations in bold have a *p* value < 0.05. Technical farm characteristics and biosecurity variables displayed in the table are those significantly associated with the abundance of each gene in the final models. AMU = Antimicrobial usage; GMR = Geometric mean ratio; 95% CI = 95% Confidence Interval; TI = Treatment incidence.

**Table 4 antibiotics-10-00820-t004:** Multivariable associations between antimicrobial usage (AMU) and characteristics and biosecurity measures of the turkey farms and the occurrence of *E. coli* isolates from turkey faeces resistant to ampicillin, tetracycline, and ciprofloxacin.

Model Variables	AMP		TET		CIP	
	OR	[95% CI]	OR	[95% CI]	OR	[95% CI]
**AMU**						
Log_10_ TI aminopenicillins	**4.10**	**[1.37, 12.30]**				
Log_10_ TI tetracyclines			3.32	[0.75, 14.7]		
Log_10_ TI quinolones					**12.85**	**[4.00, 41.2]**
**Technical farm characteristics**						
Age of turkeys at sampling (standardized)	0.83	[0.53, 1.31]	0.74	[0.48, 1.13]		
Sampling season (ref: autumn, winter)			2.13	[0.85, 5.31]		
**Biosecurity**						
Other livestock farms present within 500 m (ref: no)	0.48	[0.19, 1.18]				
Wild birds can enter the stables (ref: no)			2.67	[0.90, 7.87]		
Different age categories of turkeys present (ref: no)			0.48	[0.19, 1.20]		
The nearest turkey farm within 500 m (ref: no)					**0.28**	**[0.11, 0.69]**

Associations in bold have a *p* < 0.05. All OR shown in the table are mutually adjusted for class specific AMU and farm characteristics/biosecurity variables for the specific column. AMU = Antimicrobial usage; OR = Odds ratio; 95% CI = 95% Confidence interval; AMP = Ampicillin; TET = Tetracycline; CIP = ciprofloxacin; TI = Treatment incidence.

## Data Availability

The data presented in this study are available upon request from the corresponding author.

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
