# Peer review of "Risk Factors for Antimicrobial Resistance in Turkey Farms: A Cross-Sectional Study in Three European Countries"

_antibiotics, 2021, doi:10.3390/antibiotics10070820_

Round 1

Reviewer 1 Report

Dear authors
Thank you for the manuscript. Please rather refer to the Microsoft(R) Word document, as it contains links and Tables and may be easier to read due to its original formatting.

****************************

The authors conducted a study on an important subject. Furthermore, it is interesting to compare different approaches in a single article as it provides a lot of information altogether. Therefore, it is acceptable that the text is rather lengthy. Throughout the manuscript, actions have been described clearly and I could follow easily in the majority of the cases. Language and style are appropriate. I acknowledge that field studies are challenging, especially when conducted in different countries. Therefore, it is also acceptable that some data is missing. Nonetheless, I have some questions and remarks. You made multiple assumptions (on sample and variable selection), but have not explained so far why you made these. A bit of clarification would facilitate to read you manuscript. For now, it sometimes leaves the impression of a convenience approach rather than neutral observation, which was probably not the case.

Major comments:

- Why did you perform metagenomics analyses on native fecal samples (i.e. the whole bacterial community) and not only on E. coli given that your study design is rather comparative than explorative? You may be comparing things here which are not really comparable.

Line 458: Why did you group macrolides and lincosamindes? Did this increase your sample size per category? It is not a usual approach, I reckon. Furthermore, what you call "corresponding" AMR is not a suitable term. I would have expected "homologous" here. But given that you combined antimicrobial drugs belonging to a different class together, I cannot think of a suitable term either. Also, as an example, you perform grouping here but show TI per class (Fig. 1). It is not wrong, but not easy to follow either. It is fine that you do not want to print TI values separately for each class and farm given the size of such a table, however, I couldn't even do the calculation if I wanted to.

-It is hard to follow why you use different approaches (line 462) for different calculations. A precise explanation is needed.

- Why did you relate these advanced, precise labor-intensive and costly laboratory analyses to AMU data which is based on farmers' memory only instead of documentation? Wouldn't scrutinizing treatment records or at least a cross check with veterinary bills have been a better alternative? For what reason you did not opt for prospective data collection, given that the turnaround in turkey production is short? It would have increased your data quality considerably and may have led to more homologous AMU-AMR findings. I see a great imbalance here.

- Whenever using mutual adjustments, how did you avoid committing the "mutual adjustment fallacy" error, also called "Table 2 fallacy"? Why did you choose implementing mutual adjustments? Is this really necessary?

- Why did you choose epidemiological cutoffs? Please provide a thorough explanation. Clinical breakpoints are an alternative (Clinical and Laboratory Standards Institute: Performance Standards for Antimicrobial Susceptibility Testing; twenty-seventh informational CLSI supplement M100-S27. Clinical and Laboratory Standards Institute; Wayne, PA; 2017; The European Committee on Antimicrobial Susceptibility Testing: Breakpoint tables for interpretation of MICs and zone diameters, version 8.1; http://www.eucast.org; 2018)

Minor comments:

- Figure S1. Red dots on green areas are almost invisible for some of us (around 11% of men and 3% of women, me included). I first didn’t understand at all what the maps was about. As a courtesy, you may want to choose another colour for that your readership gets the message concerning farm distribution.

Line 418: How many times did you visit each farm? How many flocks were sampled per farm?

Line 424: You state that farms had to meet the inclusion criteria of 3000-15000 turkey per farm. Maximal bird numbers in Table 1 are much higher.

Line 453: Did you take into account the varying bird numbers of previous breeding farms (if applicable) for TI calculation? Do you estimate that 6kg at the weight of treatment was applicable from what you saw during fieldwork? How would a lower real weight at treatment influence the outcome (see also line 93)?

Line 454: You sampled the farms approximatively one week prior to slaughter only. Why did you not calculate AMU more precisely using the real age at slaughter, thus a correct observation period length? Your AMU records (based on questionnaires) might be incomplete to an unknown extent. Therefore, you must explain why you did not use more precise cornerstones here.

Line 461/Citation 30: This citations does not longer exist. Please provide the correct document source. I do not understand your approach concerning combination products. The document I accessed found was this one (Defined daily doses for animals (DDDvet) and defined course doses for animals (DCDvet) (europa.eu)) and the ESVAC provides DDD values for combination products. Did you use these?

Line 471: Check spelling.

Line 472: Did you ship samples at -80 °C?

Line 508: Why did you choose this number? Why didn't you include all sample like in the other analysis? I do not understand why you introduce another methodology here.

Line 517: Did the amount of bacterial DNA vary largely between samples? Besides, please do not start sentences with abbreviation (throughout the manuscript).

Line 529f: I had to read it thrice and still do not understand. Please rephrase.

Line 561: What does "standardization of age" mean? Why did you do it?

Line 569: Did all categorical variable have only two levels from the beginning? Is "dichotomized" the correct term for non-continuous variables (keyword cutoff-value)?

Line 579: I can't think of what a standardized cluster would be. What do you mean? Please rephrase for more clarity.

Line 580: Given the length of your manuscript you should provide a brief explanation of what can be found in those papers and how it relates to this study? Why do you mention meta-analyses here?

Line 589: What is a gene level? The gene count; the prevalence?

Line 590: You convert 5 datapoints per farm into one. Does this affect convergence of the model?

Line 595: Add a sentence of how the R package works, if this gives any information on the variable selection process.

Line 602: Why GMR? I have a hard time following all your assumptions and adaptions. Clarification is needed as for now I have the impression that you choose convenient assumptions to detect associations of AMU with AMR. By the way, association are "observed" or "detected". I find it a rather unsuitable formulation to claim that you "found" them (Line 336).  

Line 607: These are not classes! Also, mind correct punctuation.

Line 606: This paragraph is not well structured. Name dependent variables. Precise predictor variable selection if this is what you mean in Line 607. How did you choose the cutoff ofsix farms? Mind the journal's rules for number spelling from one to twelve (if applicable).

Line 610: Why this? On EUVSEC plates, nalidixic acid is tested as well. Why did you not combine these, as they are both fluroquinolones? Once again it seems that you did it at your convenience.  

Line 614: Elsewhere in the manuscript you claim that due to uniform sampling you achieve non-country-specific risk factors (line 407). Here you include random effects for countries. Is this contradictory?

Line 616: Why didn't you use backward elimination as you did in point 4.8.3.?

Line 625: Compare to line 512. Do you use "gene" and "cluster" interchangeably? Did you target clusters or single genes in qPCR analyses?

Line 628: It is confusing. What does contribution mean here?

Line 110: How can "biosecurity status" be dichotomized? Which question do the positive answers you mention refer to? You mean all questions from Suppl. Mat. B? I think I know what you mean but being precise here would allow for speeded-up reading.

Table 1/general question: Why didn't you look at mortality? Do you consider this parameter as biologically not relevant in turkey production?

Line 158: I cannot find the information on whether you looked for AMR genes relative to the antimicrobial classes you chose to be of relevance. Could you please elicit this? Compare to line 607 and the amount of classes you used there for analysis (three). Why didn’t you use the same amount throughout the manuscript=

Line 171: Rephrase. Find a more scientific term for "positive direction". Are you sure you want to comment on non-significant results? Mind correct punctuation.

Line 190: Why did you decide to assign these values?

Line 196: Did you test whether ermB is more prevalent than the other ones?

All tables and figures must be interpretable without referring to the text. Provide corresponding captions.

Line 259: How did you decide that the prevalences were high enough?

Line 270: What does separate testing mean? Why did you do it?

Table3/4: Why did you show OR only once. Outcome variables are categorical in both analyses, right?

Line 290: "The abundance of the ermB cluster...accounted for 69.0% of the macrolide… resistance classes". This is not clear to me. Please rephrase.

Line 301: You name either bacterial species in brackets or the technique in both cases. Please be more concise.

Line 315: You mix genes and antimicrobial classes. Please rephrase for more precision.

Line 323: Is it possible to select for transmission? I don't think so.

Line 340: Do you have any suggestions why this was the case? In general, in the discussion section you shall not repeat the results but tell what the mean to us (and the turkeys).

Line 352: check spelling.

Lines 367: You already stated this elsewhere, however, you do not provide reasons for the different ways of sampling.

Lines 371: How can variation in abundance be due to a low concentration?

Lines 401: You have to decide whether you want to compare resistome and single-species results. Your statements are contradictory (see line 376).

Author Response

The authors would like to thank you for the helpful and detailed feedback. We have addressed the reviewers’ comments point by point in the attachment.

Reviewer 2 Report

 The manuscript   entitled “Risk Factors for Antimicrobial Resistance in Turkey Farms: A 2 Cross-Sectional Study in Three European Countries” talks about the identification of risk factors for antimicrobial resistance (AMR) in turkey farms in three European countries (Germany, France and Spain) between 2014 and 2016.Overall, the study is clear and concise. The introduction is relevant and theory based. Sufficient information about the present study rationale and procedures are provided for the readers. The methods are generally appropriate, although clarification of a few details are required. Overall, the results are clear and compelling. The authors make a systematic contribution to the research literature in this area of investigation particularly when the study is expanded to three different countries.

Specific comments follow.

  • There is abundant literature in the file of antimicrobial resistance and some in the fields of farms too. Why the previous study findings are not included in the introduction.
  • Please make the materials and methods more clear and fluent. It could be difficult to follow for the readers if it gets published in the way it is.
  • I am not quite sure that all the data in the supplementary files is directly related to this study or performed by the authors. Please be care full to add any data which is not original to this study or the description of which is not available in the materials and methods.
  • As far as I understand the study was done in Germany, France and Spain. Is there any reason the countries are referred by alphabets B E H.
  • Can authors add the answers to the questionnaire at Supplementary Materials Part B: Selected biosecurity check questions from the questionnaire used in risk factor analyses.
  • The authors should mild the tone of the paper at various points. For example Line 405, “One of the strengths of our study” could be reformed as “The study is important from the fact that”. Similarly Line 413“Nevertheless, we can still state” can also be reformed in a milder tone. Line 372” presence of inhibition” could be inhibition of gene expression”

Author Response

The authors would like to thank you for the helpful feedback. We have addressed the reviewers’ comments point by point in the attachment.

Reviewer 3 Report

The present study, entitled „Risk Factors for Antimicrobial Resistance in Turkey Farms: A 2 Cross-Sectional Study in Three European Countries” concerns an extremely important topic. The research issue undertaken by the Authors is highly justified due to the growing problem of microbial resistance to antibiotics in both humans and animals. This is a huge public health problem, causing a significant health and economic burden worldwide. Antibiotic resistance is spreading rapidly and unpredictably, especially since the types of antimicrobials used in food-producing animals are often the same or closely related to those used in human medicine. One possible source of antibiotic resistance transmission is turkeys and turkey meat, especially since turkey fattening is the second largest meat production sector after broiler production.

The results of the presented study have of great importance for both science and veterinary practice. The aim of the paper is clearly formulated. Sampling from three different European countries makes the results more reliable and of higher applicability value. The manuscript presents the topics in an orderly and logical manner. The title and abstract correspond to the content of the paper. The layout of the work and its internal structure is appropriate. The value of the work is increased by the use of modern methods of genes identification. The selection of sources and literature is current and complete.

In the opinion of the reviewer, the paper lacks general information about the mechanisms of antimicrobial resistance in E. coli (for example ESBL which is one of the most clinically and epidemiologically important mechanisms of drug resistance in Enterobacteriaceae - it is based on the production of β-lactamases with an extended substrate spectrum, which have the ability to hydrolyze penicillins, cephalosporins (with the exception of cephamycins) and monobactams, with the most important being their activity against III and IV generation cephalosporins; AmpC cephalosporinase production, carbapenemase production (MBL) and other).

Author Response

The authors would like to thank you for the helpful feedback. We have addressed the reviewer’s comments in the attachment.

Reviewer 4 Report

Dear authors,

This is a solid study, with some important findings regarding AMU and AMR in turkey farms in EU and suitable for publication in this journal.

I have only some minor questions/issues:

  • Table 1: Median weight of turkeys in country E is 0.1? With a range 0.1-0.5, is this right?
  • Line 112: Change "statusses" with "statuses"
  • Figure 2. I am a little confused regarding the AMR gene abundance. You have sampled 20 turkey farms from each country, but you present results regarding AMR gene abundance in metagenomics from 21 farms in country B and 19 from country H. Please explain.
  • Line 321: Do you agree that a possible explanation regarding the generally stable composition of AMR genes in all samples has probably to do with the intrinsic presence of AMR genes in various non-pathogenic bacteria of the gut microbiota? If yes, you may add a comment about this in this section.
  • If I understood correctly, only one farm reported use of aminoglycosides. So, there was probably no real pressure regarding the prevalence of aph3’-III gene. It seems, according to your results, that lower contact with birds/vermins increases aph3'-iii gene prevalence and higher contact lowers it, regardless of aminoglycosides use. Do you agree with this, and if so can you add a comment on this?
  • I suggest you could add a comment about the various numbers of 16srRNA gene copies in the genome of the various bacteria species that may interfere with the normalization results when comparing results from different farms and/or countries where the gut microbiome may be different e.g. because of diet differences etc.

Author Response

The authors would like to thank you for the helpful feedback. We have addressed the reviewer’s comments point by point in the attachment.
